



# A benchmark data set for long-term monitoring in the eLTER site Gesäuse-Johnsbachtal

Florian Lippl[1], Alexander Maringer[3], Margit Kurka[1], Jakob Abermann[1], Wolfgang Schöner[1], and Manuela Hirschmugl[1,2]

[1]Department of Geography and Regional Sciences, University of Graz, 8010 Graz, Austria
[2]Joanneum Research Forschungsgesellschaft mbH, DIGITAL–Institute for Digital Technologies, 8010 Graz, Austria
[3]Nationalpark Gesäuse, 8913 Admont, Austria

**Correspondence:** Florian Lippl (florian.lippl@uni-graz.at)

**Abstract.** This paper gives an overview over all currently available data sets for the European Long-term Ecosystem Research (eLTER) monitoring site Gesäuse-Johnsbachtal. The site is part of the eLTSER platform Eisenwurzen in the Alps of the province of Styria, Austria. It contains both protected (National Park Gesäuse) and non-protected areas (Johnsbachtal). Although the main research focus of the eLTER monitoring site Gesäuse-Johnsbachtal is on inland surface running waters,

forests and other wooded land, the eLTER whole system (WAILS) approach was followed in regard to the data selection, systematically screening all available data in regard to its suitability as eLTER's Standard Observations (SOs). Thus, data from all system strata was included, incorporating Geosphere, Atmosphere, Hydrosphere, Biosphere and Sociosphere. In the WAILS approach these SOs are key data for a whole system approach towards long term ecosystem research. Altogether, 54 data sets have been collected for the eLTER monitoring site Gesäuse-Johnsbachtal and included in the Dynamical Ecological Informa-

tion Management System – Site and Data Registry (DEIMS-SDR), which is the eLTER data platform. The presented work provides all these data sets through dedicated data repositories for FAIR use. This paper gives an overview on all compiled data sets and their main properties. Additionally, the available data is evaluated in a concluding gap analysis with regard to the needed observation data according to WAILS, followed by an outlook on how to fill these gaps.

## 1 Introduction

The past decades show an unprecedented change in biodiversity and ecosystem functions and services. Direct drivers (e.g. land and sea use changes, direct exploitation of organisms, climate change) and indirect drivers (e.g. production and consumption patterns, human population dynamics and trends) influence this accelerating change (Pecl et al., 2017; Díaz et al., 2019; Malhi et al., 2020). In order to get at a better understanding of natural variability and human induced changes in biodiversity and ecosystems, long-term monitoring systems with open accessible and harmonized data at a broad spatial and temporal scale

are of crucial importance (Shin et al., 2020). Loescher et al. (2022) establish the Global Ecosystem Research Infrastructure Network, an independent and site based research infrastructure, addressing and tackling future global ecosystem challenges. On a European scale the infrastructure is called European Long-Term Ecosystem Research (eLTER). The goal of this paper is twofold and follows the concept of the whole system (WAILS) approach (Mirtl et al., 2021) by focusing on eLTER's

Standard Observations (SO) from all system strata (Geo-, Atmo-, Hydro-, Bio- and Sociosphere) and considers the FAIR
(Findable, Accessible, Interoperable, Reusable) data principle (Wilkinson et al., 2016). Depending on the site category different criteria have to be met in order to realize the holistic approach. Site categories refer to the quality of the measurements done on site. Category 1 is the highest site category where all strata have to be covered and and SOs in at least two strata need to be measured with advanced methods (prime method). Category 2 also demands coverage of all strata, however, simpler measurement methods of the SOs can be applied (basic method). The method type also refers to the temporal and spatial
resolution of the measurement. More details can be found in Mirtl et al. (2015) where they describe the eLTER network and framework with the focus on Austria as well as in Zacharias et al. (2021) and Mirtl (2022). It is noteworthy that categories A (very high priority) and B (high priority, but needed for further discussion), as described in Zacharias et al. (2021) and Mirtl (2022) have meanwhile been replaced by the already mentioned categories 1 and 2. The essential differentiation is still valid. Category 3 sites are those where data is measured via remote sensing. Further, the measured site properties are described on
the Dynamic Ecological Information Management System – Site and Dataset Registry (DEIMS-SDR), a web portal where all information about sites as well as documentation of their linked data can be found (Wohner et al., 2019).

The first aim is to give a comprehensive overview of the data currently available for the site, present them in a structured manner and provide links to the respective data repositories. The second aim is to identify remaining data gaps as well as chances and risks for future (long-term) observations.

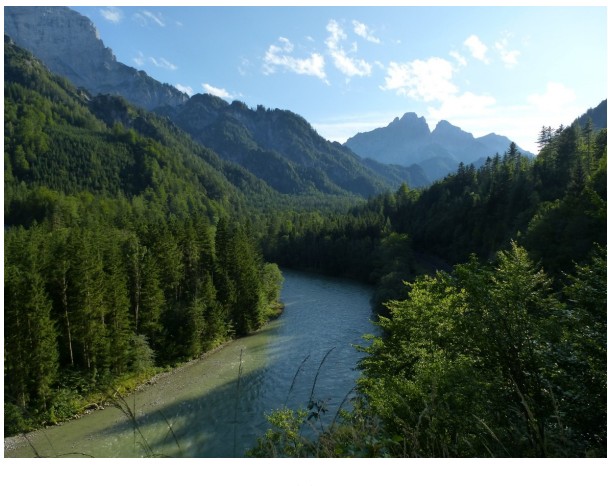
(a)

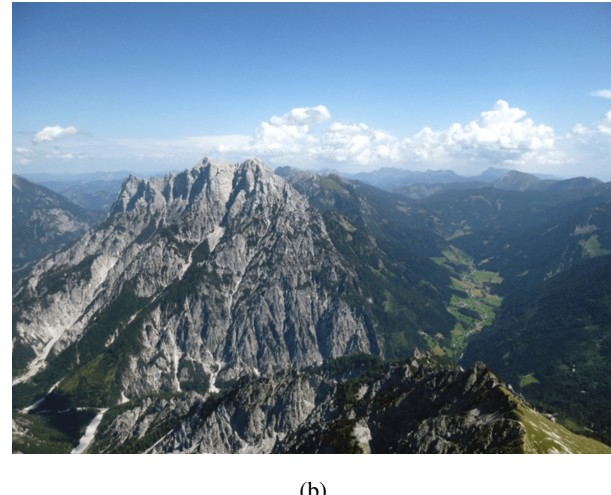
(b)

**Figure 1.** (a) Gesäuse with Enns river and view on Admonter Reichenstein and (b) from Admonter Reichenstein, with the Hochtor mountain range on the left and the Johnsbachtal on the right (photographs by G. Lieb).



## 2 Site description


In the beginning of 2023 the eLTER site Gesäuse-Johnsbachtal was formed. It consists of two previously established separate sites (see Figure 1a): National Park Gesäuse (Maringer and Kreiner, 2016) and the Johnsbachtal (i.e. Johnsbach valley, Strasser et al. (2013)). As part of the of the eLTSER platform Eisenwurzen only the southern part of the National Park Gesäuse is integrated in the newly merged site. However, the whole original Johnsbachtal catchment is included to the new site. Figure 2

illustrates the delineation of the new eLTER site, including the stations and their operators. Gesäuse-Johnsbachtal politically belongs to the province of Styria in Austria (see map in Fig. 2). The Gesäuse-Johnsbachtal is part of the Ennstal Alps, composed of carbonate and crystalline rocks (Strasser et al., 2013).

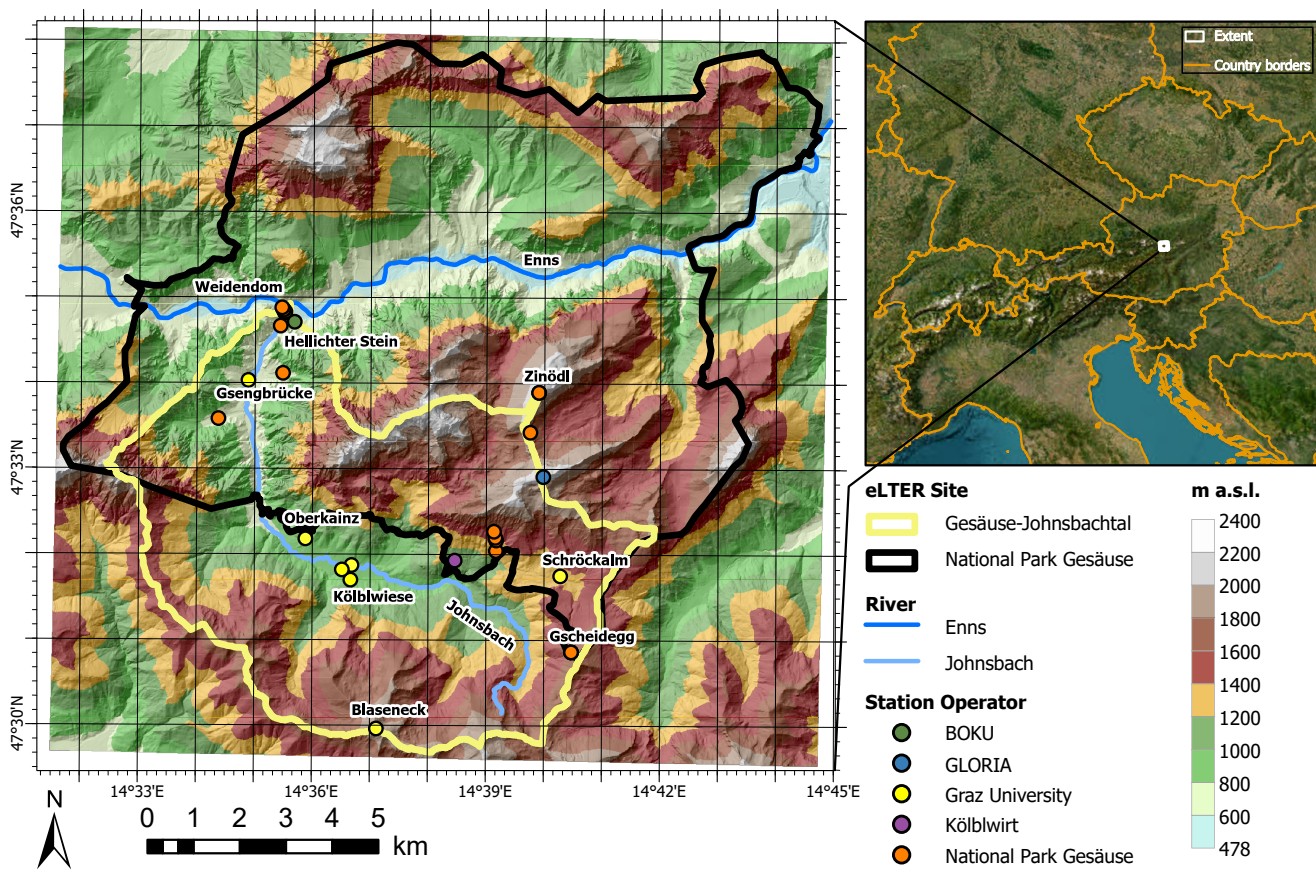

**Figure 2.** Study site overview (left) including the different measurement stations (background: basemap from Esri (2009)) and country borders (right, from Eurostat (2020)).



The region is characterized by a steep mountainous landscape intersected by the Johnsbach creek in the central part of the site (see Fig. 1b). The Eisenerz Alps form the boundary to the south-east. The total area of the Gesäuse-Johnsbachtal site 50 covers approximately 155 km², with significant variation in elevation and high relief energy, which is a dominating factor for the natural processes in the area. While the Johnsbachtal valley is at an altitude of 600-700 m a.s.l., the highest elevations are found in the area of the summit of the Hochtor (2369 m a.s.l.). Due to the great range of altitudes within a small area, the Gesäuse-Johnsbachtal encompasses extremely diverse habitats and, consequently, species of animals and plants. Climate conditions are characterized by annual mean air temperatures of about 8 °C in the valley floor and about -2 °C to -1 °C in the 55 highest summit regions and with annual precipitation amounts of approximately between 1300 mm to more than 2000 mm for the same elevation range (Wakonigg, 2012a, b). In general, the study region is dominated by mountain forests along with high Alpine rock formations and meadows. These complex topographic, hydrological, geological, geomorphological and meteorological conditions pose a scientific challenge for all kinds of environmental monitoring and modelling. In addition, the combination of protected areas in the north and unprotected areas in the south makes the site particularly interesting, as the 60 impact of human intervention on nature can be studied under similar conditions.

## 3 Data compilation

The data selection followed the comprehensive list of eLTER SOs (see Zacharias et al. (2021) for more information). Thus, the compilation includes a collection of previously existing data from the individual eLTER sites National Park Gesäuse and Johnsbachtal (e.g. climate data) and was complemented with a large variety of data from new initiatives and measurements 65 within the scientific and operational ecosystem monitoring in the study area. We structure all collected data according to the already mentioned system strata:

– Geosphere: observations regarding geological and pedological properties;

– Atmosphere: atmospheric observations (temperature, wind, precipitation, etc.);

– Hydrosphere: hydrological observations (runoff, water temperatures, snow cover, etc.);

– Biosphere: observations with regards to fauna and flora;

– Sociosphere: selected parameters regarding economy and society;

Within each stratum, there are several SOs. A SO can either comprise only one variable (e.g. leaf area index (LAI) of forests on site scale, see Table 3) or is defined by a bundle of variables (e.g. meteorological data). These variables are called comprised SO variables. In Tables 2 to 6 we list all SOs and all variables with existing data for the Gesäuse-Johnsbachtal site. Important 75 interlinkages and interdependencies exist among these parameters. One example is the anthropogenic influence on sediment discharge of Johnsbach river (Strasser et al., 2013). Due to the abandonment of a former quarry, the sediment regimes have changed significantly. It is therefore of great importance to include all strata in the long-term observation plan. Generally, data presented in this work is published under the CC-BY license.





## 3.1 Geosphere data

Most of the geological and pedological SO information stems from a project called FORSITE (see Table 1), which was carried out in the province of Styria, Austria, in 2021, based on a standardized data collection and analysis approach (Land Steiermark, 2022; Klebinder et al., 2022; Winkler and Wilhelmy, 2022). Spatially inclusive and comprehensive raster data indicating various properties of the geological substrate (Land Steiermark, 2022; Winkler and Wilhelmy, 2022) and soil (Land Steiermark, 2022; Klebinder et al., 2022) overlying bedrock was derived from thousands of field observation sites. The accuracy of the

raster data is dependent on the underlying geological and pedological information and the different methodologies used. More details on accuracies can be found in (Winkler and Wilhelmy, 2022; Klebinder et al., 2022; Land Steiermark, 2022). In case of the geological substrate map properties (GIS Steiermark, 2021b) information was derived from existing geological and pedological information from pre-FORSITE projects and literature (see Winkler and Wilhelmy (2022), Klebinder et al. (2022), https://www.geologie.ac.at/en/shop/maps and https://www.data.gv.at/application/digitale-bodenkarte-ebod/) as well as

geological mapping results from the FORSITE project, including forest road mapping with detailed description of more than 2800 points. Additionally, 240 representative samples were tested in the laboratory for geological and mineralogical properties (Winkler and Wilhelmy, 2022). The geological substrate map considers homogeneous mapping units greater than 1 ha or 50 m x 100 m (Land Steiermark, 2022). In case of soil (GIS Steiermark, 2021a), nutrient and water infiltration rate (GIS Steiermark, 2021c, d) information was derived from pre-existing soil data combined with newly mapped data during during the FORSITE

project, including 1800 field points, of which 400 were tested in the laboratory (Klebinder et al., 2022). As described in detail in Klebinder et al. (2022) and Land Steiermark (2022) the collected data, combined with available climate, topography, geology (geological substrate) and remote sensing data, was analyzed using an artificial neural network algorithm resulting in the available raster data (GIS Steiermark, 2021a, a, c, d). Accuracy attributes are not shown on the publicly available data (GIS Steiermark, 2021b), but can be obtained from GIS Steiermark upon request. Detailed information about the FORSITE project,

the different approaches and the background of available data can be found in Land Steiermark (2022); Klebinder et al. (2022); Winkler and Wilhelmy (2022).

Table 2 illustrates how the pedological and geological FORSITE data relate to the eLTER Geosphere SOs. All pedological and geological data are deducted from the FORSITE project parameters, except for the variables on geological site characterization by Bauer et al. (2018), who distinguish 11 distinct classes and the variable soil type classification by Carli (2007).

Not included in the eLTER SO catalogue, but also part of the Johnsbachtal monitoring system, are bedload transport measurements conducted by the University of Natural Resources and Life Science Vienna (Habersack et al., 2017; Kreisler et al., 2017; Rascher et al., 2018). The bedload transport is measured near the bridge close to Weidendom (see the red marker in Fig. 2) via a geophone, which is calibrated by using bedload traps and basket samplers.

## 3.2 Atmosphere and Hydrosphere data

Most of the atmospheric and hydrospheric variables included in the eLTER SOs are continuously measured as part of the WegenerNet (see variables with "/WEGC/" in doi string) by the Wegener Center for Climate and Global Change of the Univer-





sity of Graz. The WegenerNet Johnsbachtal comprises 13 meteorological stations distributed over different altitudes and one hydrological station (see Fig. 2), with 6 different station operators (Fuchsberger et al., 2021). The network concept, metadata information and first analysis have been published in a separate data paper (Fuchsberger et al., 2021). Data is made available
via the data portal of the WegenerNet (https://wegenernet.org/portal/jbt/) where they are explained in detail. Therefore, we solely give a brief summary of the measured data listed in Tables 3 and 4 for completeness. Also sensor specifications with the corresponding accuracies are available on the portal. Here, we focus on the most important sensors. The sensors used for relative humidity measurements have an error margin of $\pm\,2\,\%$ within a relative humidity of $0\,\%$ to $90\,\%$. Above $90\,\%$ the margin increases to $\pm\,3\,\%$. For air temperature the accuracy is $\pm\,0.2\,°C$ at $20\,°C°$. The error in precipitation measurements ranges from
$-\,1\,\%$ to $+\,1\,\%$ for both unheated and heated sensors. Wind direction is measured with an average error of $\pm\,2\,°$ at a wind speed of $12\,m\,s^{-1}$. Air pressure measurements show an accuracy of $0.15\,hPa$. Water level is obtain with a precision of $\pm\,2\,mm$. The global radiation measurements show a non-linearity and a tilt error of less than $1\,\%$, a temperature dependent sensitivity of less than $4\,\%$ and a directional error smaller than $20\,Wm^{-2}$.

**Table 1.** FORSITE data.

| Variable | Description | Source |
|---|---|---|
| soil texture | based on the soil texture triangle in ÖNORM L 1050 indicating the soil type (e. g. sand, silt, loam, clay, clayey sand...) based on the sand, silt, clay content of the soil | GIS Steiermark (2021a) |
| soil bulk density | mean density of the mineral soil layer | GIS Steiermark (2021a) |
| soil pH | classes of soil acidity (pH CaCl2) of mineralogical soils, inherently indicating the availability of nutrients | GIS Steiermark (2021c) |
| total organic C concentration | concentration of organic carbon | GIS Steiermark (2021c) |
| CEC (cation exchange capacity) total nitrogen | total concentration of nitrogen | GIS Steiermark (2021c) |
| soil base saturation | base saturation of the mineralogical soil | GIS Steiermark (2021c) |
| particle size distribution | particle size distribution of the geological substrate (upper layer), with focus on the matrix material (fine material), distinguishing between coarse material (g) and fine material (f+: clay, silt, f: silty, clayey sand, f-: sand) | Winkler and Wilhelmy (2022) |
| soil infiltration rate | average percolation capacity of organic (humus) and mineralogical soil combined with the subsoil substrate layers | GIS Steiermark (2021d) |

Water level data from the station Weidendom (see Fig. 2) from the WegenerNet are complemented by data from the station
Gsengbrücke. Discharge at Gsengbrücke at the lower part of Johnsbachtal has been observed since 2011 with water level and flow velocities recorded automatically. Manually discharge measurements using different methods, depending on runoff amounts, complement the automated measurements for establishing rating curves. A total of 28 discharge measurements were taken for rating curve estimation between 2011 and 2020, performed by the Hydrographic Service of Styria. In addition, a stage with markers is mounted at the side of the station. Observations and data indicate that high flow velocities change the river bed





**Table 2.** Variables - Geosphere.

| SO | Comprised SO variable | eLTER SO code | Unit | DOI |
|---|---|---|---|---|
| | soil texture | SOGEO_001 | - | https://doi.org/10.23728/b2share.26e0f6a0d23d4b75aee1a6ca6402c802 |
| soil inventory | soil bulk density | SOGEO_001 | $g\,cm^{-3}$ | https://doi.org/10.23728/b2share.7d7ed01e4ff44cf288b4787fcd8c29db |
| - | soil pH | SOGEO_001 | -log(H+) | https://doi.org/10.23728/b2share.1828cbfff216412eb7fa080df0cd340b |
| pedological/geological characterization | soil type classification | SOGEO_001 | - | https://doi.org/10.23728/b2share.dc9c169d8de14f79b5d679332846ea82 |
| | geological site characterization | SOGEO_001 | - | https://doi.org/10.23728/b2share.184f861d5217412aaff0ab0d98e8a052 |
| soil chemical | total organic C concentration | SOGEO_003 | $g\,kg^{-1}$ | https://doi.org/10.23728/b2share.693e3536a5d546b299d4118712656bbf |
| and | CEC total nitrogen | SOGEO_003 | $g\,kg^{-1}$ | https://doi.org/10.23728/b2share.a9460dc767454d3ea061e41115812a02 |
| physical characteristics | soil base saturation | SOGEO_003 | % | https://doi.org/10.23728/b2share.cbce5e8962244e68a99e31d012ca6bc4 |
| percolation/infiltration - soil | infiltration rate soil | SOGEO_048 | $mm\,d^{-1}$ | https://doi.org/10.23728/b2share.01fa335e7e9349c49c10a0d9db97e945 |
| sediment (aquatic and marine) inventory | particle size distribution | SOGEO_155 | - | https://doi.org/10.23728/b2share.9a43891693c946a8bc99d7a6481649e2 |

considerably. Those high water amounts typically occur during snowmelt or heavy precipitation events. Usually, comparing recorded water level values with manual discharge measurements allow for an estimate of a stage-discharge relation. However, due to major changes in the river bed, we apply two different relations: one between 01-01-2014 and 20-06-2017 (Qh1) and another one between 01-01-2019 and 25-01-2021 (Qh2). The measurement before 2014 and between autumn 2017 and early 2019 show large discrepancies between stage values recorded with the ultrasonic sensor and the stages with markers. Therefore,

data between 20-06-2017 and 01-01-2019 was omitted. In the lack of more information on low flow conditions we remove all calculated values below the lowest measured discharge value (0.585 $m^3\,s^{-1}$). Figure 3 shows the estimate of discharge based on the different stage-discharge relations for the period 2014 to 2021. A clear seasonal cycle is visible that relates to snowmelt and heavy precipitation events in summer. From a rough estimate, we estimate the accuracy of discharge is below $1\,m^3\,s^{-1}$, but may be higher for extreme events/values. We note that the values differ from Seier et al. (2020) (their Figure 3) as they

used uncorrected values stored in the logger and did not account for major reworking in the river bed between 2017 and 2019. The last two variables from the Atmosphere and Hydrosphere not part of the WegenerNet are vegetation phenology and LAI. Vegetation phenology is a product of the Copernicus Land Monitoring Service High-Resolution Vegetation Phenology and Productivity suite (ESA, 2021). The data includes information such as start date, end date and peak of the season. The detailed validation process of the following summary is outlined in Camacho et al. (2021). The validation includes a comparison

of satellite data with ground measurements. However, reference data is sparse. Therefore, an intercomparison with related products is conducted which facilitates a validation over larger areas or greater time periods. In general, the validation assesses temporal and spatial consistency as well as goodness of fit of two products. The LAI is derived from Sentinel-3/OLCI data. It has a spatial resolution of 300 m. Its validation follows the guidelines of the CEOS Land Production Validation group. In the validation process three benchmark datasets are used for intercomparison and one global product for direct comparison. The

accuracy assessment is updated continuously as part of the quality monitoring. Based on Martínez-Sánchez (2022) the LAI product is operational and further information on accuracy can be found in Fuster et al. (2020).



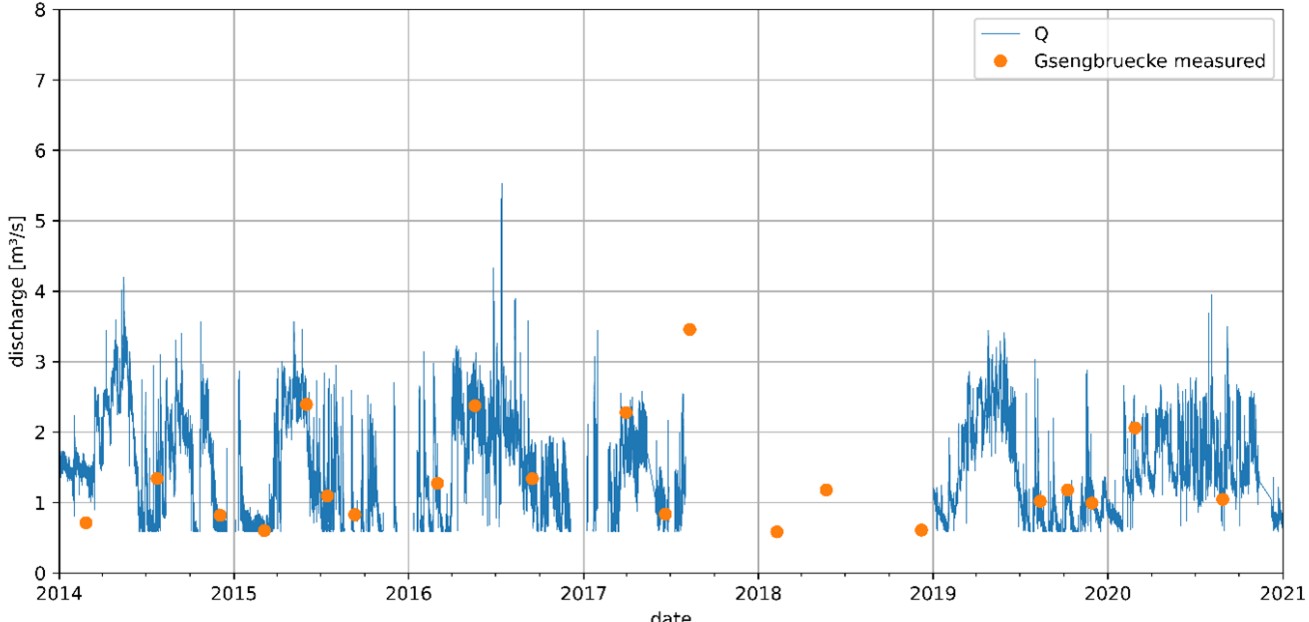

**Figure 3.** Discharge at Gsengbrücke (blue line) based on two different stage/discharge relations. The orange dots mark discharge measurements done with different methods. The data gap between autumn 2017 and 2019 is related to strong changes in the riverbed.

**Table 3.** Variables - Atmosphere.

| SO | Comprised SO variable | eLTER SO code | Unit | DOI |
|---|---|---|---|---|
| | relative air humidity | SOATM_027 | % | https://doi.org/10.25364/WEGC/WPS8.0:2023.2 |
| | precipitation | SOATM_027 | mm | https://doi.org/10.25364/WEGC/WPS8.0:2023.2 |
| meteorological data | air temperature | SOATM_027 | °C | https://doi.org/10.25364/WEGC/WPS8.0:2023.2 |
| | wind speed/direction | SOATM_027 | m s$^{-1}$, ° | https://doi.org/10.25364/WEGC/WPS8.0:2023.2 |
| | surface atmospheric pressure | SOATM_027 | hPa | https://doi.org/10.25364/WEGC/WPS8.0:2023.2 |
| radiation | global radiation (incoming and reflected) | SOATM_028 | W m$^{-2}$ | https://doi.org/10.25364/WEGC/WPS8.0:2023.2 |
| vegetation phenology (site scale) | phenological traits (including start, maximum and end of season) | SOBIO_016 | date | https://doi.org/10.23728/b2share.08a00dc24fc9450d8ea96b94d7825915 |
| | leaf area index - forests (site scale) | SOBIO_025 | index | https://doi.org/10.23728/b2share.8371e67b90604d44b15c7268edc17670 |
| | leaf area index - non forested sites | SOBIO_026 | index | https://doi.org/10.23728/b2share.8371e67b90604d44b15c7268edc17670 |

## 3.3 Biosphere data

Most of the biosphere variable data (see Table 5) has been extracted from the database of the National Park Gesäuse via the BioOffice software (Zimmermann, 2010). This was done for the SOs vegetation composition and birds, bats, frogs and
insects. To obtain the vegetation composition three permanent plots representing highly dynamic habitats (avalanche chutes, gravel streams) are used. Vegetation plots were mapped by locating triangulation points for permanent traceability in this harsh



**Table 4.** Variables - Hydrosphere.

| SO | Comprised SO variable | eLTER SO code | Unit | DOI |
|---|---|---|---|---|
| physical and chemical water characteristics - surface water (running waters) | water temperature | SOHYD_005 | °C | https://doi.org/10.25364/WEGC/WPS8.0:2023.2 |
| water level - surface water (running water) | water level | SOHYD_010 | m | https://doi.org/10.25364/WEGC/WPS8.0:2023.2 https://doi.org/10.23728/b2share.5cee0a54ee7c49b095a65a13e90d964e) |
| snow cover and depths | snow cover | SOHYD_012 | cm | https://doi.org/10.25364/WEGC/WPS8.0:2023.2 |
| | snow depth | SOHYD_012 | cm | https://doi.org/10.25364/WEGC/WPS8.0:2023.2 |
| soil water content/soil temperature | soil water content | SOHYD_168 | - | https://doi.org/10.25364/WEGC/WPS8.0:2023.2 |
| | soil temperature | SOHYD_168 | °C | https://doi.org/10.25364/WEGC/WPS8.0:2023.2 |

environment (Klipp and Suen, 2011). In future there will be consistent surveys precisely at these three locations also striving to incorporate other species groups. Acoustic sampling, camera trapping and soil sampling is feasible in these permanent plots and will be tested on-site as soon as possible. Considering the fish communities, in the framework of the LIFE-project (Haseke,

2010) conservation activities and structural improvements were made in the Johnsbach river. The impact of these actions on the Johnsbach river was assessed by measuring the fish population. All details can be found in Fischer and Gumpinger (2015). Further, in 2022, a study about the ecological state of the Johnsbach river was conducted (Bernatz and Gauer, 2022) where both macroinvertebrate and macrophyte communities were analyzed at three locations. For quality checks the national park applies its own tests to ensure data integrity including plausibility checks of coordinates, biological taxa and accordance

of monitoring methods (Maringer and Kreiner, 2021). Inconsistencies are highlighted by computer-based rules and checked manually. Regular re-use of data in projects and for management purposes is especially helpful to evaluate and improve data quality. Commissioned work is provided under FAIR principles and CC-BY or similar licences, unless protected species or habitats where immediate release is not intended.

**Table 5.** Variables - Biosphere.

| SO | Comprised SO variable | eLTER SO code | Unit | DOI |
|---|---|---|---|---|
| vegetation composition (mainly species level+abundance) | ground vegetation | SOBIO_017 | descriptive | https://doi.org/10.23728/b2share.646a96773ba944808d67421bb7ab1b26 |
| birds, bats, frogs, insects using acoustic recording | | SOBIO_018 | number | https://doi.org/10.23728/b2share.dbd12671044c4022a267b8515655b7ee |
| fish community - running waters | | SOBIO_083 | number | https://doi.org/10.23728/b2share.56707952146643548e33bd5f11d82ae5 |
| macrophyte community (quantitative) - freshwater, transitional water | | SOBIO_086 | % | https://doi.org/10.23728/b2share.d970b611e9b145d0a14a0b87be504ad5 |
| macroinvertebrate community (quantitative) - running waters | | SOBIO_181 | individual m$^{-2}$ | https://doi.org/10.23728/b2share.06877c87da0b462e9e2b5e0f76afd874 |

### 3.4 Sociosphere data

In the sociosphere, all SOs are composed by one variable, except status of employment, as illustrated in Table 6. Many of the sociosphere SOs are qualitative-descriptive SOs. This means that there is a report rather than a quantitative data set in



the background. The SO area under tillage is the area and amount of land regularly ploughed, the basic data is taken from the INVEKOS data base BML and AMA (2022) and currently covers the years 2015-2022. The SO governance structure and character gives insight into the municipality board, the municipality council members and the political characteristics of the
area. The SO stakeholder engagement provides information on the effectiveness of the processes within an organization or project. The SO basic services provision: health and education describes schools, health and social welfare. NUTS 3 (nomenclature of territorial units for statistics) and LAU (local administrative units) represent the administrative subdivision of the region, such as municipalities and communes, to which the site belongs. The SO age profile-education, attainment-residential, profile-residential density gives a demographic and socio-economic overview of the Gesäuse-Johnsbachtal for the years from
2019 to 2023. The SO status of employment is divided into different age groups and gender. The SO extraction of minerals for the Gesäuse-Johnsbachtal site focuses on avalanches and mass movements in form of rockfall, debris flows and mudflows. The SO protected area displays the extent of the site which belongs to the National Park Gesäuse and the protected landscape "NSG1" of the Johnsbachtal.

**Table 6.** Variables - Sociosphere.

| SO | Comprised SO variable | eLTER SO code | Unit | DOI |
|---|---|---|---|---|
| | area under tillage | SOSOC_029 | % | https://doi.org/10.23728/b2share.e1e1a5fc9bbd426b80c521668da3ffac |
| | governance structure and character | SOSOC_032 | descriptive | https://doi.org/10.23728/b2share.d79f860da5e644f79aa976f8604fbacf |
| | stakeholder engagement process indicators | SOSOC_033 | descriptive | https://doi.org/10.23728/b2share.af2570112ba5411483c5c35e8dd0824a |
| | basic services provision: health and education | SOSOC_034 | descriptive | https://doi.org/10.23728/b2share.68a8d3bb3d984d4dbfe9ad38b392553d |
| | NUTS3 and Local Administrative Units (LAU) spatial databases | SOSOC_041 | map | https://doi.org/10.23728/b2share.996d672f2d904bedb7b2a4d9ccc8f706 |
| | age profile-education, attainment-residential, profile-residential density | SOSOC_043 | descriptive | https://b2share.eudat.eu/records/566f3ca5d09b463d8834938b74f838a3 |
| status of employment | employment (employment rate %; employment by sector; unemployment) | SOSOC_044 | % | https://doi.org/10.23728/b2share.e47f8b05476d47fea098df66db6c1761 |
| | extraction of minerals | SOSOC_150 | - | |
| | protected areas | SOSOC_153 | map | https://doi.org/10.23728/b2share.95330101354a42be8c8a3aa45e6ff013 |

# 4 Gap analysis, risks and chances

## 4.1 Gap analysis

A large amount and variability of data has been collected so far. Figure 4 shows how many SOs and their related variables are already covered in the Gesäuse-Johnsbachtal site considering, all system strata and site categories. In our analysis, the criterion for a SO to be considered as obtained is that at least one of its variables is measured. We can see, that Geosphere and Sociosphere are currently well covered (in terms of SOs >50 %), while the other three strata show coverages of only about
25 %. The covered strata are more or less in line with previous assessments done for all other eLTER sites (Mollenhauer et al., 2018). Although, Mollenhauer et al. (2018) used components instead of the currently used strata, there is still a large similarity on what is measured elsewhere compared to our site (see Figure 3 of Mollenhauer et al. (2018)). In their component water





budget for example, discharge has been measured in 24 % of the sites. For their component abiotic heterogeneity, (in essence weather data, air temperature, windspeed and soil characterization) more than half of the sites had measurements (61 % for air temperature, 60 % for wind speed and 51 % for soil). These are equivalent to our atmospheric and geospheric variables. However, since the focus of the Gesäuse-Johnsbachtal site lies on the habitat of inland surface running waters and forest targeting the eLTER category 2 standards, the number of required SOs and variables has been reduced (see Fig. 5). This results in a coverage of more than 50 % for the required SOs besides the Biosphere. Nevertheless, less than 50 % of the SO variables are measured.

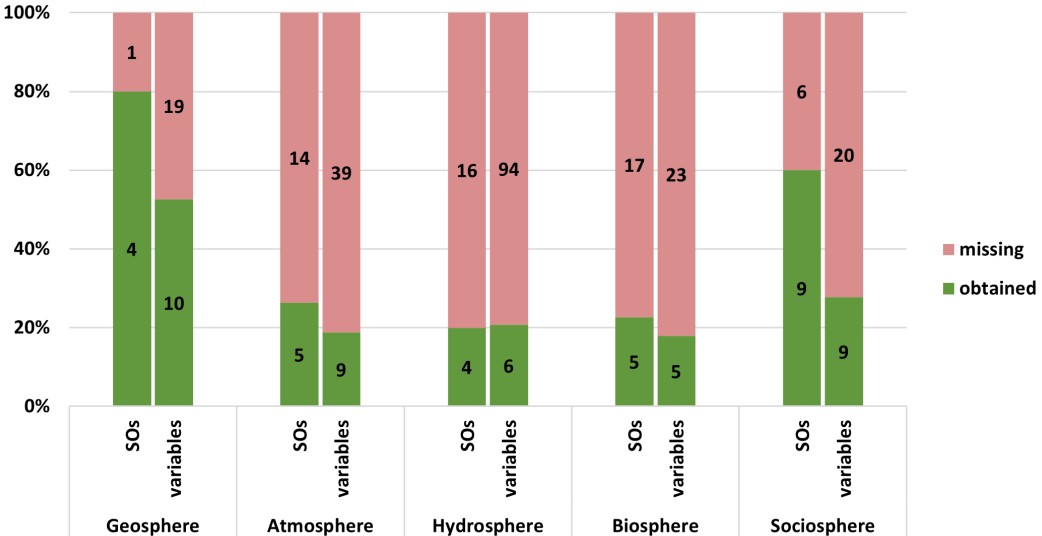

**Figure 4.** Current coverage and gaps in terms of all standard observations for the Gesäuse-Johnsbachtal site.

## 4.2 Risks and chances

In addition to the incomplete SO and heterogeneous coverage, a potential limitation is the temporal resolution of some data sets. While the atmospheric parameters are measured every 10 minutes, the other parameters have varying temporal resolution.

However, most of the geospheric information is considered as background data, which by nature only requires one assessment (disregarding geological timescales). For Biosphere, so far assessments have been done without following specific protocols (now under discussion). Thus, some surveys have been done only once. Establishing long-term monitoring gained momentum since the national park was founded but is still in its early stages. The missing repetitive measurements for some parameters are a risk for the long-term monitoring system. Due to limited funds, not all measurements have been given the priorities needed for a consistent long-term monitoring under the WAILS regime so far. Socio-ecological studies have been done once in a while covering e.g. landscape history (Schafferhofer, 1998; Hasitschka, 2014) and sustainable tourism (Obenaus, 2005; Dockhorn, 2021) without taking SOs into account. Provided that there is a fruitful exchange between the different

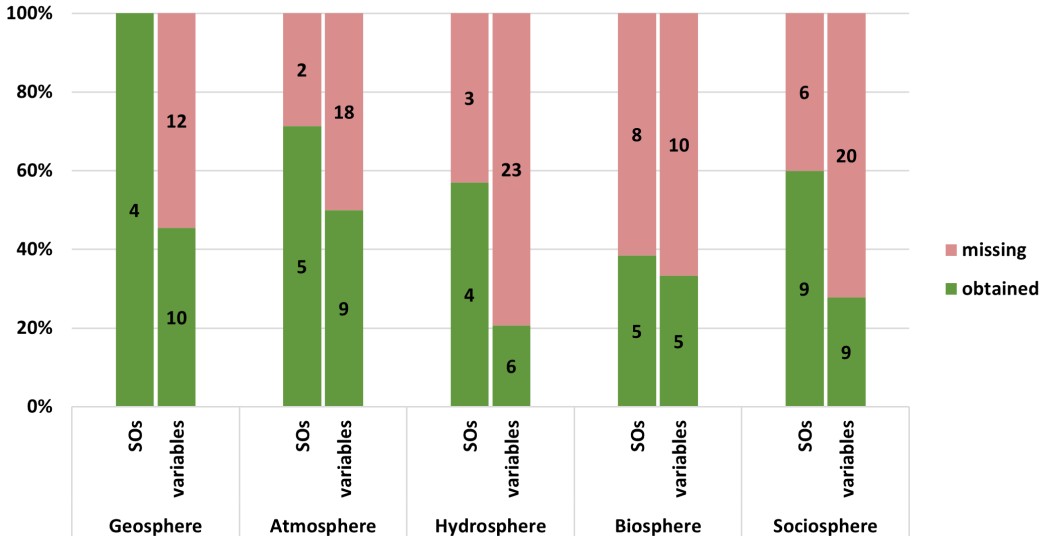

**Figure 5.** Current coverage and gaps in terms of the standard observations for inland running waters and forests (category 2) for the Gesäuse-Johnsbachtal site.

research institutions active at the site and that there is a drive to provide compatible data, there is a substantial chance for future advances. This also calls for a mutual effort to gain sufficient funding for the purpose of leveraging the full potential of the data for a better understanding of natural processes, mainly driven by climate change in this vulnerable mountain environment.

## 5 Conclusions

This study gives a comprehensive overview of the data currently available for the eLTER site Gesäuse-Johnsbachtal in terms of standard observations and entailed SO variables. We present them in a structured manner and provide links to the respective data repositories to facilitate data finding and uptake for future research and applications. The identification of remaining data gaps revealed that in spite of many efforts, there are still significant data gaps. Naturally, these gaps are more significant when considering all SOs in the WAILS approach than when only considering the SOs of the thematic focus on inland running waters

and forests. However, more efforts should be made to further measures to provide as many SOs as possible in this standardized manner and to provide them in a FAIR way. We consider the approach outlined in this paper as an opportunity to function as a showcase for other sites, where lots of data is available, but scattered and not easy to find. Therefore, eLTER and the European research infrastructure activities are a huge chance to leverage the full potential of the long-term observations already available and to complement them with new observations still.



*Data availability.* Data described in this article can be accessed at https://b2share.eudat.eu/, https://www.parcs.at/npg/ and https://wegenernet.org/portal/. The temporal resolution of the published data varies. Detailed information about the temporal resolution, metadata and download links of the data can be found at the DEIMS Data and Site Registry (Maringer et al., 2023). Data will be continuously updated and added to the aforementioned data portals, however, it will be published under a new version with a new DOI.

*Author contributions.* Conceptualisation, A.M., M.H.; methodology, F.L., A.M., M.H.; formal analysis, F.L., A.M., M.H.; investigation, all;
data curation, F.L.; writing—original draft preparation, F.L. A.M., M.K., J.A., M.H.; writing—review and editing, all; visualisation, F.L., J.A.; supervision, M.H.; project administration, M.H.; funding acquisition, M.H. All authors have read and agreed to the published version of the manuscript.

*Competing interests.* The authors declare that they have no conflict of interest.

*Acknowledgements.* This project received funding from the Earth System Sciences funding program of the Austrian Academy of Sciences.
Originaltext lt. Vertrag: Das Projekt wird aus Mitteln des Earth System Sciences Förderprogramms der Österreichischen Akademie der Wissenschaften finanziert



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
