# Peer review of "A benchmark data set for long-term monitoring in the eLTER site Gesäuse-Johnsbachtal"

_Earth System Science Data, 2024_

## Referee Comment (RC1)

Review ESSD 2024-12, Gesause-Johnsbachtal eLTER data

Nice effort by authors to compile and document Gesause-Johnsbachtal eLTER data but fails to meet ESSD expectations.

ESSD attempts to provide "a wide range of openly accessible, high-quality, well-documented and highly useful data products" (https://doi.org/10.5194/essd-10-2275-2018). This collection might, with modification, meet 'documentation' expectations but misses entirely on 'accessible', 'quality', and 'useful' criteria. Manuscript needs major revisions or authors should file a .pdf report elsewhere.

One could meet all FAIR expectations (free, accessible, interoperable, reusable) with garbage! FAIR misses entirely indications of quality, utility, etc. Readers would like to believe that these authors compiled quality data products but we get no indications. If they focused on "inland surface running waters, forests and other wooded land", one would hope that these data would prove of high enough quality and utility to complement (or contradict) global or European streamflow data, global or European forest cover data (e.g. as provided by FAO?), or global or European land use data (e.g. clearance of woodland for agriculture, reversion of former agriculture to woodland, managed vs unmanaged woodlands, etc.). Unfortunately, reader gets nothing. (Many relevant data sets already exist in ESSD; not hard for authors to find them.)

Reader also confronts only barriers in trying to access data. Links in paragraph starting at line 225 fail completely. (Most of these apparently exist under CC-BY-NC licenses, also verboten for ESSD.) Link at DEIMS, for which reader needs an intuitive search, cover only the park. Access fails across the board. Again, quoting ESSD guidelines (above): users "should enjoy fast free reliable "two-click" access: one click to a relevant landing page and a second click to download." This reviewer appreciates challenges and difficulties in providing seamless access to so many data products from so many differing sources, but without good efforts readers will not develop confidence in this particular group of authors nor in this particular compilation.

Apparently eLTER data from this site (except perhaps slowly varying geophysical data) meet (most often) fewer than 50% of authors own expectations for temporal and spatial coverage. A few sentences about 'doing better' (e.g. lines 212, 213) will likely have minimal impact. How does this site compare to other eLTER sites? Does the entire eLTER network face similar challenges? If this represents a unique compilation, does it also represent uniquely good or uniquely bad coverage? Not the responsbility of these authors to assess the entire eLTER network/effort, but they must show outcomes of the G-J networks as useful, helpful, reliable. Nothing in current text assures readers/users? Show us skills, accomplishments, etc. If you feel you have a worthy data product to share, prove it!

A database of European soil properties exists, described and published in ESSD (https://doi.org/10.5194/essd-15-4417-2023). If soils represent one of the strongest (so far) outcomes of this effort, reassure us by showing how these data compliment, fill gaps in, or contradict the larger European data product. Same for any other variable? E.g. Austrian met service maintains records of temperature, humidity, wind, precip in some of these locations. How do these eLTER data fit, or not fit, those external data? Authors may protest: not their job to do all these inter-comparisons. But if not this group, who? And, if these data do not somehow add to, fill gaps in, or otherwise compliment existing mountain, Austrian, central European, or Europe-wide data products, what use to establish an eLTER and what use to publish them?

If the Gesause-Johnsbachtal eLTER efforts provides quality-certified potentially-useful data, as this reviewer believes, and if authors wish to publicize and provide that data to a wide

community via ESSD, then they need major rewrite to meet ESSD standards (note, not weaker FAIR standards). They may present readers with a necessarily-hesitant but overall positive start, but, with current incomplete description, they have not helped readers to recognize those facts nor authors' skills as data providers.

Summary:

1) Assure easy reliable access to all data: quick easy and without restriction!
2) Convince readers of utility.
3) Clearly convey - where available - known uncertainties.
4) Show relevance of these data to other eLTER or biosphere sites (e.g. Spanish mountain sites already published in ESSD; Uni Bern phenology data, also high elevation and also already published in ESSD).
5) Prove to readers that this group has made best efforts at validation.
6) Read and meet as much as possible ESSD guidelines (at https://www.earth-syst-sci-data.net/10/2275/2018/).